# SECDFAN: A Cyber Threat Intelligence System for Discussion Forums Utilization

**Georgios Sakellariou ***, **Panagiotis Fouliras** and **Ioannis Mavridis**

Department of Applied Informatics, School of Information Sciences, University of Macedonia,
54636 Thessaloniki, Greece
* Correspondence: geosakel@uom.edu.gr; Tel.: +30-2310-88-2850

**Abstract:** Cyber Threat intelligence (CTI) systems offer new capabilities in the arsenal of information security experts, who can explore new sources of data that were partially exploited during the past decades. This paper deals with the exploitation of discussion forums as a source of raw data for a cyber threat intelligence process. Specifically, it analyzes the discussion forums' characteristics and investigates their relationship with CTI. It proposes a semantic schema for the representation of data collected from discussion forums. Then, it applies a systematic methodology to design the reference architecture of the SECDFAN system, which handles the creation of CTI products following a comprehensive approach from the source selection to CTI product sharing and security experts' collaboration. The final product of this work is the SECDFAN reference architecture. The contribution of this paper is the development of a CTI reference architecture of a system that, by design, handles all CTI-related issues for creating CTI products by analyzing the content of discussion forums.

**Keywords:** cyber threat intelligence; information security; system modeling; discussion forums; reference architecture

## 1. Introduction

In today's cybersecurity landscape, threats such as zero-day attacks, advanced persistent threats (ATPs), and ransomware are risks to which most organizations are exposed daily. Those threats are so complex in nature that they take traditional detection and response security systems to the limits. Moreover, organizations with prevention capabilities are most likely to be the winners of the cybersecurity battle [1]. One of the cornerstones of prevention capabilities establishment is cyber threat intelligence (CTI).

CTI is the domain of cybersecurity where data collected from various sources are analyzed and assessed regarding threat actors and their motivation, the methodology of a cyberattack, and the victim, aiming to help the defenders to detect, prevent, or predict a cyberattack by providing indicators related to the stages that comprise a cyberattack [2]. Overall, a major scope of CTI is to provide security experts with the possible ways that an attack will occur.

This paper focuses on the utilization of discussion forums from CTI systems for creating CTI products [2]. Our motivation originates from the fact that the CTI systems proposed in the bibliography deal with certain but not all aspects of the CTI process [2] that utilizes discussion forums.

The research goal of this paper is to systematically design a CTI system capable of generating CTI products from open or secret communities via data collected from discussion forums on the surface Web, deepnet, and the darknet. Toward this goal, we set the following research questions:

- What are the characteristics of those raw CTI data and their sources?
- Which components comprise a CTI system that uses those raw CTI data?
- How can we model this system?

To answer those research questions, we design and propose the SECurity Discussion Forums ANalysis (SECDFAN) system, which models in detail the production and sharing of CTI products in a standard format from raw CTI data collected from discussion forums. At the same time, the design methodology followed provides an in-depth analysis of the CTI challenges related to discussion forums. The reference architecture of the SECDFAN system constitutes the main contribution of this paper since it addresses, by design, all issues related to the creation of CTI products by employing a content analysis of discussion forums.

More specifically, in Section 2, a fourfold review of related works is presented. We outline the works on CTI system modeling (Section 2.1), discuss CTI extraction from text sources (Section 2.2), discuss works of computer-mediated conversations (CMC) and discussion forums (DF) analysis (Section 2.3), and highlight the limitations of the works dealing with CTI systems for discussion forums (Section 2.4). Section 3 presents the design methodology that we have applied in the case of SECDFAN. In Section 4, we analyze the characteristics of computed-mediated conversations and discussion forums to reveal those that CTI can exploit. Finally, in Section 5, the design steps and the development of the SECDFAN are presented and analyzed in depth. A conclusion and future work are presented in Section 6.

## 2. Related Work

To set the base of this paper, we discuss the related works of the four pillars of this research, separately. Specifically, we discuss: (a) CTI system modeling, (b) threat intelligence extraction of text data, (c) computer-mediated conversations and discussion forums, and (d) CTI systems for discussion forums.

### 2.1. CTI System Modeling

The systematic design and modeling of systems have been studied in many works related to system engineering [3]. However, few papers exist that focus on the development of CTI systems [4–6] and fewer on the systematic design and modeling of a system [2].

Jo et al. [6] designed a system for automatically extracting threat intelligence from unstructured text data. However, they mainly focused on the extraction technique and not on the modeling of the system following a systematic approach.

Wagner et al. [4] presented the design of a widely used threat intelligence sharing platform (MISP); however, that project took place in the early stages of the CTI domain, and the authors did not follow a systematic modeling methodology.

Similarly, Cha et al. [5] designed a blockchain-based CTI system that ensured sustainability, but they did not apply a systematic modeling methodology.

Finally, in [2], we proposed a reference model for CTI systems and a methodology for designing those systems, which we used in this work to design SECDFAN.

### 2.2. Threat Intelligence Extraction from Text Data

Although a vast quantity of cybersecurity-related data are expressed in pure text, there is no significant work on the automatic processing of those sources in the context of a CTI process [2]. However, in most of the existing publications, semantic techniques, natural language processing (NLP), graph mining, and other machine-learning techniques have been adopted for the automatic processing of text sources. Wang et al. [7], for example, extracted threat intelligence data from unstructured security articles and reported using semantic techniques.

In Motoyama et al. [8], data collected from six underground forums were analyzed to derive their structural and qualitative characteristics, to determine social network relationships and user communication, to identify types of goods and services traded, and to estimate the effect of social status on user interaction. However, nothing related to CTI was discussed.

Basher et al. [9] proposed a latent Dirichlet allocation (LDA) model-based approach to extract crime-relevant topics, as well as topic–author relations and activities from chat logs

analysis. Moreover, they proposed metrics and criminal topic models which adapted the extensive research of topic detection to the needs of forensics. Nevertheless, the extracted results were not used in combination with CTI processes.

Liao et al. [10] published one of the first works in the field of CTI in which natural language processing (NLP) was utilized for extracting indicators of compromise (IoC) from technical articles collected by crawling open security blogs. Their results were expressed in the OpenIoC format, an open standard during the article's publication, which limited the results on specific types of indicators. Although its bias for OpenIoC was not in accordance with current industry trends, this work set the basis for the automatic extraction of CTI from text data.

Grisham et al. [11] based their research on neural networks (NN) and social network analysis (SNA) to identify mobile malware from hacking forums and relate it with malware authors. Moreover, they tried to evaluate the social characteristics of malware authors and their influence in forums. Towards this goal, they followed a typical collection–preprocessing–analysis methodology, but they did not clearly state the collection mechanism. In addition, the results of their analysis were not expressed in any of the CTI standards (e.g., STIX v2 [12]), even though the authors stated that they were willing to involve Malware Attribution Enumeration and Characterization (MAEC) [13] in the future.

Finally, Husari [14] proposed a text mining technique, which involved the combination of NLP and information retrieval (IR) techniques, to extract threat actions from unstructured security reports. Additionally, a model that combined extracted actions into attack patterns was proposed, while results were expressed in the widely accepted STIX [12] standard.

### 2.3. Computer-Mediated Conversations and Discussion Forums

Computer-mediated conversations (CMC) have been meticulously studied, and conversation analysis has been used in social sciences, health, and education for various scopes in many publications. Paulus et al. [15] presented a comprehensive review of conversation analysis techniques and applications.

Herring [16] discussed CMC in-depth and defined them as any textual exchange between two or more participants via information technologies such as email, instant messaging, real-time chat, and discussion forums.

Uthus et al. [17] analyzed chats and the applicable analysis techniques and methodologies such as chat preprocessing, chat room feature processing, thread disentanglement, topic detection, message attribute identification, user profiling, social phenomenon detection, and automatic summarization.

Holtz et al. [18] determined that discussion forums were a type of asynchronous textual communication among community members. They also analyzed their tree structure and identified the structural similarities and the overall differences (i.e., permitted text length, synchronous communication) between discussion forums and real-time chats.

Finally, Hoogenveen et al. [19] analyzed the tree structure of discussion forums determining that conversations corresponded to threads, which were logically separated into topic-oriented sections. They used this observation for web forums' semantic and text analysis.

### 2.4. CTI Systems for Discussion Forums

Collecting and analyzing data from websites in the darknet, deepnet, and surface web for cybersecurity is the subject of many works. Here, we present several of them and discuss their limitations and issues, which motivated us in this work. Liao et al. [10] proposed the iACE solution for automatic extraction of indicators of compromise (IoCs) from security articles published in blogs. However, they only dealt with some of the CTI challenges (e.g., source selection, stealthiness) and they used only the content of the articles, not the information regarding the bloggers themselves.

Li et al. [20] developed the NEDetector system to extract neologisms from hacking forums. However, they mainly focused on the analytic parts of the system and did not handle issues related, for example, to data collection and selection of the sources.

Jo et al. [6] proposed the Vulcan CTI system for extracting CTI data from unstructured text. However, they focused only on the analytics part of their system and the semantic processing of the text.

Similarly, Deliu et al. [21] designed a CTI system in the form of a model that automatically searched hackers' posts and extracted CTI. However, again, they mainly focused on the analytics part of their model and did not provide details, for example, of the quality measurement.

Koloveas et al. [22] proposed the INTIME, a machine-learning-based framework that combined several tools and services in one architecture to support security analysts in dealing with various data sources in the process of developing CTI products. However, the proposed architecture was not based on a reference model. Consequently, several gaps appeared in how INTIME handled some CTI-related issues, such as the quality of the refined CTI products.

Finally, Sapienza et al. [23] proposed a CTI system that used data from cybersecurity experts and forums to generate CTI and to alert cybersecurity specialists of an imminent attack. However, the designers of the proposed CTI systems focused only on specific components which dealt with specific CTI issues, such as analytics.

To summarize, we observed a general tendency for the architecture of the proposed CTI systems to focus only on specific parts and CTI-related issues, resulting from the fact that these authors did not follow a reference model during their design. This limitation motivated us to develop a reference architecture for systems that process discussion forums data as a source of CTI, by utilizing the reference model we proposed in [2].

## 3. Methodology

In this paper, we developed the SECDFAN's CTI reference architecture by following the cyber threat intelligence reference model (see Figure 1a) and the requirements analysis method (see Figure 1b) that we proposed in [2].

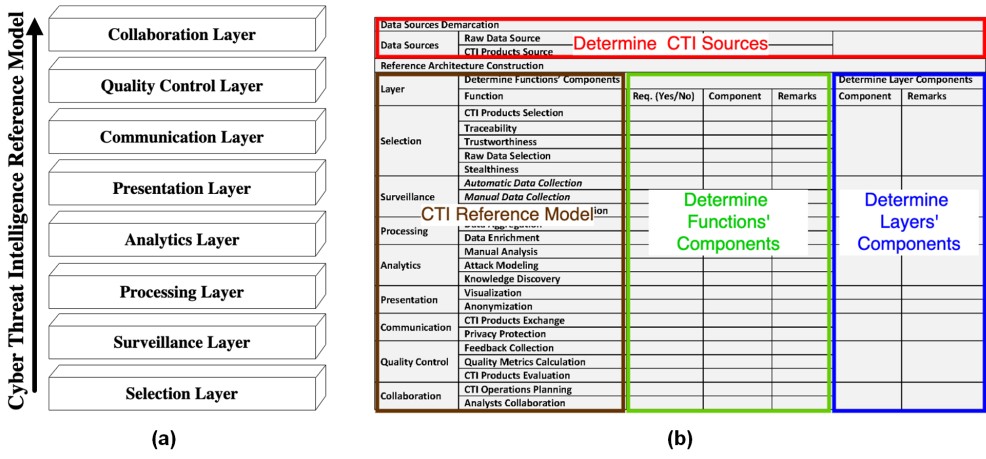

**Figure 1.** CTI: (**a**) Reference Model, (**b**) Reference Model Application Matrix.

The CTI reference model (see Figure 1a comprises eight *layers*, each dealing with a critical capability of a CTI system, specified in turn by a set of *functions* (see Figure 1b), with information following a **bottom-up** direction. Therefore, a CTI system's reference architecture is a combination of layer's components that handle the generic capabilities of the layers, and function components handling the specific functions of a layer such as as those proposed by the CTI reference model [2]. The *selection* layer handles the capability of the CTI source selection based on its unique characteristics (e.g., type of data, quality of data, source trustworthiness). The *surveillance* layer refers to the data collection capability of

a CTI system. The *processing* layer refers to the processing capability (e.g., data attribution, feature extraction, correlation, normalization) of a CTI system on the collected data that prepares them for analysis. The *analytics* layer refers to the capability of a CTI system to transform the processed data into exploitable CTI information. The *presentation* layer refers to the capability of a CTI system to either transform the analysis results into CTI products or to visualize them. The *communication* layer handles the communication of the resulting CTI products between the CTI system and other systems. The *quality control* layer refers to the capability of a CTI system to deal with quality issues of CTI products. Finally, the *collaboration* layer handles the capability of a CTI system to support and manage CTI operations.

## 4. Analysis of Discussion Forums

Any textual exchange between two or more participants carried out via information technologies is characterized as CMC [16]. This paper focuses on analyzing data collected from a subset of CMC, discussion forums (DFs).

Discussion forums are a type of asynchronous textual communication among community members. They have many similarities in their structure with real-time chats although they differ in the permitted text length and the fact that a chat is synchronous communication [17]. Therefore, data processing techniques and methodologies widely used in chat analytics can be utilized by CTI to analyze information gathered from cybersecurity-related discussion forums.

To leverage these techniques for the benefit of CTI, we analyze the structure of discussion forums and their participants' roles, we introduce their content linguistic characteristics, and we present a comparison of DFs based on their source.

### 4.1. Structure and Roles

Discussion forums follow a tree structure [18,19] in which conversations correspond to threads, logically separated into topic-oriented sections. A common discussion forum structure is depicted in Figure 2.

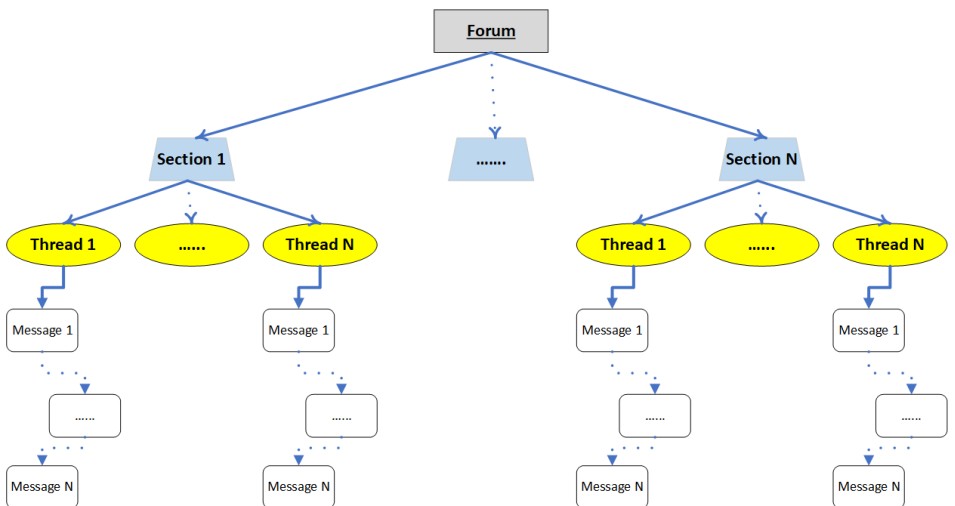

**Figure 2.** Common Discussion Forum Structure.

### 4.1.1. Discussion Forums' Content Structure

In CTI, the form of the data and any relationship which exists between the subjects under examination are important and valuable. In the case of DFs, participants and discussions are investigated by analyzing the structure of threads and messages to identify the abstract characteristics and relationships that are of interest to CTI.

In DFs, a unique identifier, which usually takes the form of a *username*, is assigned to each user and is created during registration. A user's main activity is communicating with other members of the forums community by posting messages on threads. A *message*

comprises three parts [17]: a *username*, a *timestamp*, and the text data. The combination of *username* and *timestamp* plays the role of the message's signature, while the *username* interrelates a user with a message's text data. The message structure and sequence are depicted in Figure 3.

| Username | Timestamp | Text Data |
|---|---|---|
| B0b1 | 22-09-2019 22:10AM | Awesome product, I will buy it for sure. |
| Al1c3 | 22-09-2019 23:10AM | Thank you, see the link www.shell.shop |
| ChArl13 | 22-09-2019 22:10AM | Provide access?? |
| Al1c3 | 22-09-2019 23:10AM | Accepted |

**Figure 3.** Typical Message Structure.

Moreover, the position of a message in a thread can reveal the interaction between users participating in it. This interaction usually follows the structure of an oral conversation (response sequence), but as we further analyze in Section 4.2, the contents differ from those of oral communication. Thus, a message posted to a thread either acts as a response to the thread initiator or as a reaction to a user's message. In any case, a thread can be further analyzed as several *discussions*, which are not recognized by their content, but these discussions can be discovered by message positions and timestamps. Figure 4 depicts the typical thread structure.

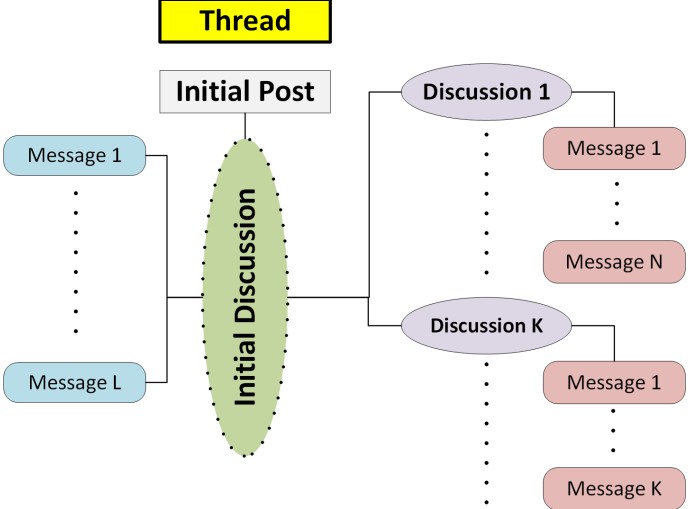

**Figure 4.** Typical Thread Structure.

To clarify the typical thread structure, we used the data collected from a real dark web discussion forum, as an example. In that case, a thread's initial post (e.g., GENERAL FEATURES: –2500 lines of code...) was from a user who was selling a new malware. This post triggered an initial discussion where other users replied directly to the initial post by posting new messages (e.g., "nice will try", "do you recommend a good crypter?"). However, in some of those replies, the thread's initiator or other users decided to reply (e.g., "well I would vouch for mine", "give it a try...") and create a new discussion. Thus, the typical thread structure can describe and organize these discussions' structure and interactions.

4.1.2. User Capabilities and Roles

The user's role and status usually depend on forum characteristics. Thus, there are forums in which users are characterized as either simple users or administrators, while other forums support a reputation-based user role system.

To model a user's role in a discussion forum, we followed a simple approach since any user reputation or characteristic of a user can be separately recorded in CTI. Additionally, at this point, we aimed to analyze the basic characteristics of conversations rather than evaluate the capabilities and the value of discussion forum participants. Therefore, in the context of CTI modeling of discussion forums, a user may participate in a discussion forum either as a common user or as an administrator.

A *common user* is any participant in a discussion forum community. Irrespective of the way access is granted to them, a common user can initiate a thread, post a message to an existing thread, delete a message of a thread, or post a reply to a specific participant on a thread.

On the other hand, an *administrator user* extends the role of a common user by adding capabilities such as user creation and management, section creation, threads management (open, close, and delete a thread), and granting access.

The role of a user and the capabilities offered to them are important in CTI when the interactions between users are modeled and when a data collection procedure is implemented. Hence, from a CTI point of view, the following user capabilities are the most significant: (a) thread initiation, (b) message posting, (c) post reply, and (d) granting access. The first three user capabilities are related to the processing and analytics layers of the CTI reference model see Figure 1a, whereas the fourth one is related to the surveillance layer.

*4.2. Linguistic Characteristics*

In the previous subsections, we analyzed the structure of DFs and the possible information which can be extracted from them. However, the main data source on DFs is the raw text data included in each message. Besides the search for usual threat intelligence such as URLs, code snippets, etc., the natural language processing (NLP) of messages can reveal techniques, tactics, capabilities, etc., related to the participants of a thread. However, when NLP processes a message, the unique message language characteristics affect the overall process and the expected results. Hence, any system that processes discussion forum messages should be aware of the linguistic peculiarities of those messages in comparison with other forms of formal or informal text communication.

In the bibliography, linguistic research [17,24–27] focuses on the study of real-time chat language. However, the main differences between DFs and real-time chat are a user's available response time and message length [17]. Thus, we can safely infer that real-time chat and DFs share the same linguistic characteristics. Based on that, the DFs' language is characterized by the following attributes:

- Abbreviations, acronyms, clipped words, and omission of subject pronouns are frequently used (e.g., "YOLO").
- Emoticons and nicknames abbreviations are common practice (e.g., "LOLcat").
- Stripping vowels is widely used to reduce the number of keystrokes (e.g., "Hll wrd").
- Capitalization, spelling, and punctuation are used to mime oral characteristics such as emphasis (e.g., "YEAh").
- Numbers and letters are combined to replace longer words (e.g., "10ths")
- Absence of grammar and usage of unstructured and informal patterns (e.g., "Cost???") is common practice.

Furthermore, the asynchronous nature of DFs minimizes the side effects that text communication causes in the meaning and structure of a conversation compared to the oral equivalents expected in other forms of CMC, such as real-time chat. In addition, even though all the above compose a well-defined starting point for the analysis of the possible DF's linguistic characteristics, there is a lack of research that analyzes the linguistic characteristics of cybersecurity-related DFs.

### 4.3. Comparison of Darknet, Deepnet, and Surface Web Discussion Forums

Discussion forums, valuable for CTI analysis, are located either on the surface Web, deepnet, or darknet. The surface Web includes all websites indexed by search engines. A deepnet includes any website that cannot be indexed by search engines, such as database-driven or where a login is required [28]. A darknet includes all websites that use encryption, for example, onion routing and the Tor browser, to hide their existence from public view. In this paper, we use the term *forum source* to refer to *surface Web, deepnet*, and *darknet* when we describe the location of a website that hosts a discussion forum.

Whilst the structure of CTI-interesting discussion forums is independent of the source, the process of *access granting* to forums' participants is the distinguishing feature. This characteristic plays a significant role in CTI because it determines the data collection mechanism and the surveillance layer of the CTI reference model. To identify the characteristics of *access granting*, we studied 30 discussion forums with interesting CTI content; the authors selected these forums by utilizing publicly available sources (e.g., hacking websites, Wikipedia). Since we could not determine the dark web's size and content, we could not claim that those discussion forums were completely representative. However, this analysis offered a general impression of their characteristics which, to the best of our knowledge, had not been presented in another work. In general, an *access granting* process consists of: (a) the *initial registration phase*, (b) the *login phase*, and (c) the *thread access phase*.

In the *initial registration phase*, **83%** of forums had in place a mechanism that requires a username, password, and e-mail address to register. In more detail, **64%** of darknet forums only required a username and password, while **79%** of surface Web and deepnet forums implemented an email-based verification process. Furthermore, **1%** of forums required special qualifications during the initial registration phase, e.g., a referral mechanism, in which the recommendation of an existing forum's member was either mandatory or registration fees were required.

In the *login phase*, **83%** of forums had a simple login mechanism, whereas **20%** employed an additional CAPTCHA mechanism for bot protection. Two-way authentication was not employed by default in surface Web and deepnet forums; nevertheless, it was offered as an option in **21%** of them. In the *thread access phase*, **80%** of discussion forums provided access to the entire content after a successful login, while **13%** of forums provided access based on the level of membership (e.g., premium membership).

In Table 1, we summarize the *access granting* characteristics of discussion forums concerning their source.

**Table 1.** Discussion Forums' Characteristics.

| Source | Initial Registration Phase | | | Login Phase | | | Thread Access Phase | |
| --- | --- | --- | --- | --- | --- | --- | --- | --- |
| | Username and Password | Email Verification | Special Qualification | Simple Login | Captcha | Two-Way Authentication | Full Access | Conditional Access |
| Surface Web | 94% | 76% | 0% | 94% | 24% | 24% | 82% | 12% |
| Deepnet | 100% | 100% | 0% | 100% | 0% | 0% | 100% | 0% |
| Darknet | 64% | 9% | 18% | 64% | 18% | 0% | 73% | 18% |
| All | 83% | 53% | 7% | 83% | 20% | 13% | 80% | 13% |

## 5. Design of SECFDAN

The complication of CTI extraction from discussion forums stems from the fact that it is a combination of typical text-mining problems such as information extraction, user profiling, and topic detection. Feldman et al. [29] discussed various methodologies which used discussion forums' data to solve a particular text-mining problem.

In this work, we designed and propose SECDFAN, which can adopt and extend any of the proposed techniques taking into account the context and special requirements of CTI. SECDFAN's design followed the methodology depicted in Figure 1b.

The first step of this methodology was to determine the type of data source, which in our case was DFs, a raw data source of textual data.

### 5.1. Selection Layer

To determine the selection layer, we needed to analyze the selection requirements of SECDFAN, as presented in Table 2. Specifically, since SECDFAN's source of data was not related to CTI products, no requirement for the related functions existed (see, rows 1 to 3 in Table 2), such as those described by the CTI reference model. At the same time, we identified that a requirement for the raw data source selection existed and, especially, it could be subdivided in the selection of DFs from the surface web, deepnet, and darknet. As a result, we had three selection components, one for each area of sources (surface Web, deepnet, and darknet) and one layer component that controlled the previous three. Those components added to SECDFAN the capability to handle the connectivity requirements of each type of DF separately and to select the most appropriate among various sources. To highlight the role of those components, we need to mention a research work that dealt with the design of specialized web crawlers [28], in which the problem of automatic access granting and its respective solving strategies [30] was examined.

In addition, stealthiness is an important requirement, especially when SECDFAN deals with darkweb DFs, in which a potential detection of a collection effort may lead the owner of SECDFAN to face an attack. Therefore, the last component of this layer offers the stealthiness (e.g., anonymization) function to SECDFAN.

**Table 2.** Application of CTI Reference Model: Selection Layer.

| Function | Req. (Yes/No) | Component | Layer Component |
|---|---|---|---|
| CTI Products Selection | No | - | - |
| Traceability | No | - | - |
| Trustworthiness | No | - | - |
| Raw data selection | Yes | Surface Web selection Deepnet selection Darknet selection | Selection |
| Stealthiness | Yes | Stealth | |

### 5.2. Surveillance Layer

The surveillance layer manages the data collection of SECDFAN. Gathering data is an essential capability for SECDFAN. Therefore, following the CTI reference model, in Table 3, we needed to identify whether SECDFAN's requirements included: (a) automatic data collection, (b) manual data collection, and (c) large-volume data collection.

The nature of DFs' origin (surface, deep, and dark Web) increases the complexity of raw data collection. DFs' data can be gathered either automatically or manually. Under the automatic approach, specialized web crawlers for each source should be developed [28]. Based on the analysis of the issues coming from the DFs' characteristics and origin [30], an ideal crawler should be automatic and independent of the individual characteristics of each DF. However, in research works [31,32], crawlers are usually forum-dependent or semiautomatic. This highlights the need for a manual collection capability in SECDFAN.

Under a manual collection of raw data, a researcher creates a fake forum account, if required, and collects data from randomly selected discussion threads in HTML pages using web browser tools. SECDFAN should support both options, even though the manual collection is inefficient for gathering extensive data volumes. Still, it is ideal for research purposes, where a proof of concept and development of simple CTI products is the main

point, because it avoids all gathering access issues that an automatic solution has to deal with.

Finally, the capability of large-volume data collection handling should be mandatory for SECDFAN and be combined with the automatic collection capability.

In conclusion, all three requirements of Table 3 should be met by SEDCFAN and represented by the respective components in Figure 9. At the same time, a generic surveillance layer component is added to the reference architecture to handle the issues such as concurrent data collection from different DFs and combined data collection following a semiautomatic approach.

**Table 3.** Application of CTI Reference Model: Surveillance Layer.

| Function | Req. (Yes/No) | Component | Layer Component |
|---|---|---|---|
| Automatic data collection | Yes | Automatic collection | Surveillance |
| Manual data collection | Yes | Manual Ccollection | Surveillance |
| Large volume of data collection | Yes | Large volume collection | - |

### 5.3. Processing Layer

The processing layer deals with the data attribution, feature extraction, correlation, and normalization capabilities of a CTI system. All these capabilities are necessary for SECDFAN and are offered by the respective processing components, as depicted in Figure 9. In addition, the functions of data aggregation and enrichment, as those described by the CTI reference model, are also necessary for SECDFAN, and they are handled by the respective components in the reference architecture (see Figure 9). A summary of SECDFAN's requirements for this layer is presented in Table 4. Next, we analyze the individual components of that layer to integrate our current work on them and demonstrate how to design a CTI system after developing its reference architecture.

**Table 4.** Application of CTI Reference Model: Processing Layer.

| Function | Req. (Yes/No) | Component | Layer Component |
|---|---|---|---|
| Data aggregation | Yes | Aggregation | Processing |
| Data enrichment | Yes | Enrichment | Processing |

In summary, the processing layer components, depicted in Figure 5, transform a DF thread's data collected in HTML pages into thread structured forms (TSFs) and produce STIX v2.1 observables.

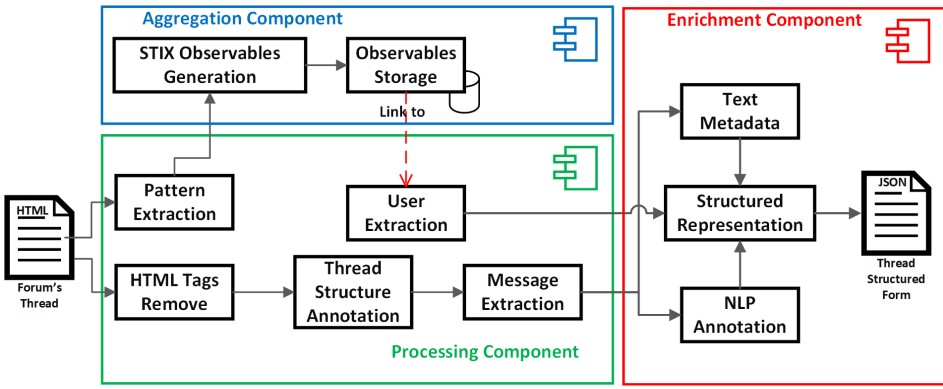

**Figure 5.** SECDFAN Processing Layer Components.

### 5.3.1. Processing Component

SECDFAN follows a processing methodology for DFs' thread data, focusing on the thread's semantic characteristics of conversation.

In the processing component, our approach focuses on extracting the semantic structure of a thread in a structural format, as described in Section 4.1.1. The processing component is fed with HTML pages, which represent an entire discussion thread. Each HTML page is processed following a pipeline, in which HTML tags are removed, an annotation takes place based on the discussion thread structure, and finally, users and messages are extracted in a semistructural format. At the same time, a set of CTI trigger-based mechanisms are applied to extract further information. A CTI trigger-based mechanism is a sequence of simple actions based on pattern discovery, which produces raw cyberobservables when specific conditions are met. For example, when a URL is identified, a domain name resolution (e.g., *lookup*) takes place. The results of those trigger-based mechanisms are fed into the aggregation component, as depicted in Figure 5.

### 5.3.2. Aggregation Component

In the aggregation component, the raw cyberobservables from the processing component are transformed into structured STIX v2.1 observables (e.g., URL, IPv4). Then, the component links the structured observables with a user participating in the specific DF-thread discussion through communication with the processing component.

### 5.3.3. Enrichment Component

The enrichment component is fed with the semistructured messages, and the users produced by the processing component. Next, the component annotates the messages for NLP purposes and text statistics (e.g., word count) are calculated and recorded as text metadata for the entire discussion thread. Finally, the component combines the extracted users, the messages, and text metadata to produce the thread structured form (TSF) described in Section 5.3.4.

### 5.3.4. Thread Structured Form and TSF Schema

A TSF is the structured JSON form in which the DF's data results after the processing layer. The TSF JSON files are the basis of the analytics layer phase and consist of thread text, relational, and NLP data, which are also populated with statistical data. The relational data are pointers that logically construct the discussions included in a DF thread.

The structure of the TSF schema follows the abstract representation of a thread. The core of the schema is the *user_object*, which represents the entity of a thread participant. The *user_object* comprises a *username* (string), a *user_id* (integer), and a bundle of *data*. The data consist of one or more *data_object*, which is an enriched representation of a message. A *data_object* comprises the *raw_data* (string), *time* (string), and *user_ref* (integer) attributes. The *raw_data* and *time* describe a message's content, while *user_ref* points to the participant to whom a message responds.

Furthermore, a unique *thread_id* plays the role of the TSF identifier, and the *dialogue* object describes the overall thread content. The *dialogue* object consists of the *forum_id* (integer), the *title* (string), and a bundle of *users*. The *forum_id* is a unique identifier of the forum from which the thread has been collected.

Finally, the attributes *classes* (integer) and *metadata* (string) complete the TSF schema. The *metadata* encapsulates any statistical data related to the thread. On the other hand, *classes* is an optional attribute that provides the researcher with the ability to classify a TSF in predefined categories. This allows an easy construction of the training datasets when machine learning is applied.

In Figure 6, the TSF JSON schema is visualized. The TSF schema is used to validate the created TSFs in the enrichment component (see Section 5.3.3).

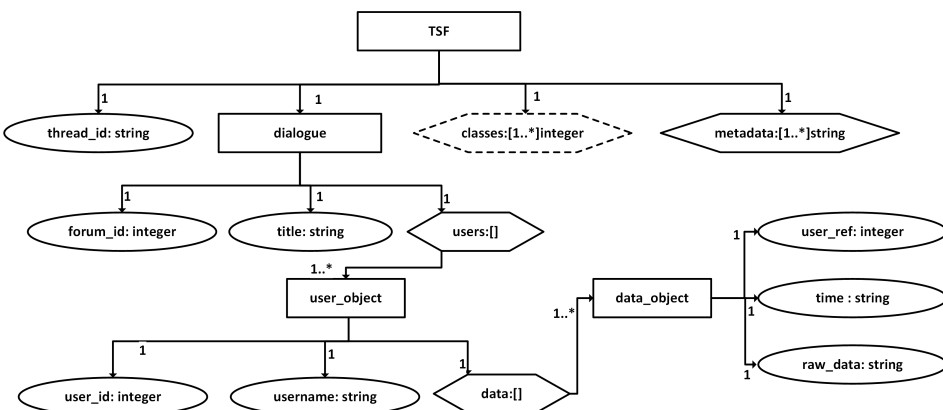

**Figure 6.** TSF Schema.

*5.4. Analytics Layer*

The analytics layer adds the intelligent part to SECDFAN. The CTI reference model determines three functions for this layer, for which we needed to determine whether a requirement exists for SECDFAN. Table 5 summarizes those requirements.

**Table 5.** Application of CTI Reference Model: Analytics Layer.

| Function | Req. (Yes/No) | Component | Layer Component |
|----------|---------------|-----------|-----------------|
| Manual analysis | Yes | Manual analysis | |
| Attack modeling | No | - | Analytics |
| Knowledge discovery | Yes | Knowledge discovery | |

In the past, data correlation techniques (such as cosine similarity) have been proposed and applied in cybersecurity text reports to quantify its significance and relevance [33]. The main objective of SECDFAN is the multidimensional analysis and correlation of data expressed in TSF format. This means a requirement for knowledge discovery does exist. Moreover, SECDFAN provides security experts with the capability to manually analyze the data, whereas no need for attack modeling exists as described in the CTI reference model [2]. In Figure 9, the components of those requirements are presented along with a layer component that aggregates their results. Next, following the approach of Section 5.3, we further analyze the analytic layer components. Figure 7 depicts the structure of the analytics layer components.

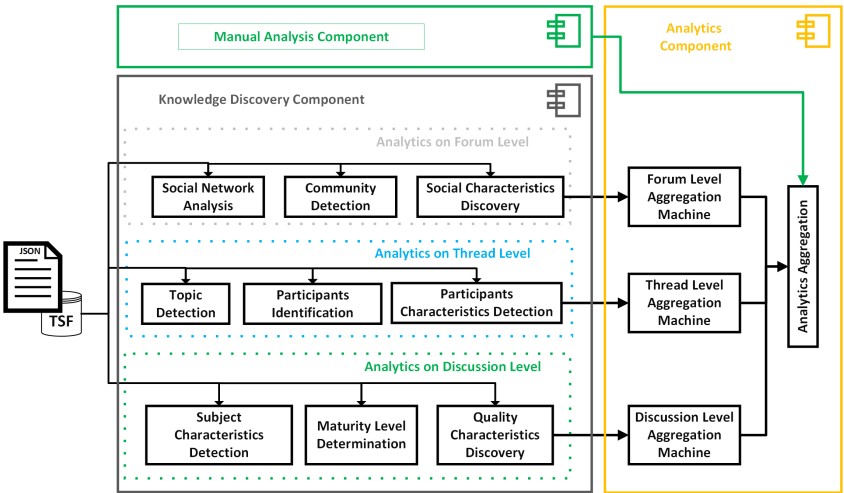

**Figure 7.** SECDFAN Analytic Layer Components.

5.4.1. Knowledge Discovery Component

The knowledge discovery component comprises three analytic levels: (a) analytics at the discussion level, (b) analytics at the thread level, and (c) analytics at the forum level. Next, we analyze those levels and their functionalities.

**Analytics at the discussion level**

The first level of the knowledge discovery component includes those analysis techniques corresponding to the context of each message. Here, the discussion messages contained in a TSF file are analyzed, aiming to reveal the characteristics of the discussion context. In particular, pattern recognition, multilabel classification, and NLP techniques are applied to each thread's message. The purpose is to:

- Detect the thread's subject characteristics (e.g., the discussed ransomware uses anti-sandbox techniques and controls local machines);
- Identify the maturity level (e.g., a bot requires advanced capabilities to be deployed);
- Determine the overall subject's quality characteristic (e.g., a keylogger is considered robust by a thread participant).

The analytics at the discussion level takes as input TSF files from which *raw_data* are extracted. Then, the techniques above are applied to these raw text data, while the whole process is supervised by the level's aggregation machine of the analytic component (see Section 5.4.2).

**Analytics at the thread level**

The thread is analyzed in depth in the second level of the knowledge discovery component. First, TSF messages are grouped and analyzed together. The grouping process follows a twofold approach. The first approach uses the messages' *time* to group them into the same order as posted in the initial thread. The second approach combines *time* and *user_ref* (see Section 5.3.4) to reconstruct discussions if they exist. The scope is to apply techniques such as latent Dirichlet allocation (LDA), authorship identification, and multilabel classifications in the thread's text data and metadata. The goals are:

- To detect the thread's topic(s) (e.g., thread A refers to a botnet, or thread B refers to the sale of ransomware);
- To identify and correlate participants' identity (e.g., participant A of thread A is the same person who participates in thread B of forum C);
- To detect participants' characteristics (e.g., participant A develops and sells malware).

This level is also applied to TSF files. Moreover, the participant's *username* is considered its identity according to [8], which states that usernames can be used as a person's identifiers among forums. This is so because participants desire unique public identities to maintain their reputations.

**Analytics at the forum level**

SECDFAN's knowledge discovery component analyzes a participant's role(s) and interaction(s) among forums and threads in the forum-level analytics. This level constructs and maintains a multigraph [34] of all forums' participants. Here, graph analysis and user classification techniques are applied, aiming to:

- Detect communities (e.g., a hacktivist group whose members exchange information about a planned DDoS attack);
- Identify participants' social network characteristics (e.g., participant A is active in forums B and C, while often participating in discussions with participant D);
- Conduct a social network analysis for forum participants (e.g., detection/recognition of trends among participants such as most participants are interested in ransomware attacks).

TSF files are also the basis for this level. However, these data are not processed once and then discarded, as in the previous two levels. There is, instead, a continuous dynamic process that updates and analyzes the multigraph.

### 5.4.2. Analytics Component

A separate aggregation machine exists in the analytics component for each of the previous levels. Those machines dynamically combine and control the functionalities of their respective levels. At the same time, this component aggregates the analysis results from the knowledge discovery and the manual analysis components. As a result, the analytics component has a twofold role: (a) to synchronize the processes of knowledge discovery component levels and (b) to accumulate the data produced.

Any knowledge discovery component level produces data, which should be fed into CTI products in the presentation layer. However, we expect those data to be asynchronously delivered and their production sequence essential. For this purpose, each level has its aggregation machine in the analytics component. Finally, the analytics aggregation machine accumulates the produced data and feeds them into the components of the presentation layer.

### 5.5. Presentation Layer

The presentation layer transforms the data of the analytics layer into CTI products. Moreover, it adds the capability of visualization to a CTI system. Here, the CTI reference model determines two functions: visualization and anonymization. In the case of SECD-FAN, the CTI products are expected to have the form of STIX v2.1 objects [35], which can also be visualized. Thus, SECDFAN's requirements include the transformation of the produced intelligence in STIX v2.1, which is represented in its reference architecture with the presentation component (see Figure 9), and the visualization of those objects, which is the role of the visualization component. Table 6 summarizes those requirements and the respective components. Anonymization is not required under SECDFAN since it uses publicly available data sources.

**Table 6.** Application of CTI Reference Model: Presentation Layer.

| Function | Req. (Yes/No) | Component | Layer Component |
|---|---|---|---|
| Visualization | Yes | Visualization | Presentation |
| Anonymization | No | - | |

Next, we analyze the presentation component and skip the visualization one because it is based on STIX objects visualization for which there are available solutions.

#### Presentation Component

The presentation component role is the creation of CTI products in the form of STIX v2.1 objects. STIX v2.1 objects are described in the *STIX v2.1 specification* [36], which is widely industry-supported, covering a large variety of CTI instances, which have extensibility and shareability as key features by being encapsulated in the *TAXII* [37] application-layer sharing protocol.

Furthermore, DFs can provide information relative to objects described in the *STIX v2.1 specification* [36]. However, the required analytics may lead to extravagant processing efforts, which is a matter of big data. To highlight the internal mechanisms of the presentation component, we focused only on threat actors, generic relationships, and observable objects, making the hypothesis that SECDFAN analyzes DF threads relative to bots, DDoS, keylogger, and ransomware, as described in the malware label open vocabulary of STIX.

Those objects are used as containers for the output of the analytics layer. Moreover, the above *STIX v2.1* objects are tailored to the requirements of DFs' analysis by using only those predefined values which are relative to the topics of *bot, DDoS, keylogger*, and *ransomware*, and are required to describe the findings of the analytics layer. The selection of properties is based on the empirical study of the forums' contents and is presented in Table 7.

**Table 7.** DFA-Specific *STIX v2.1* Objects.

| Properties | STIX v2.1 Object | |
| --- | --- | --- |
| | **Threat Actor** | **Observable** |
| Discussion-level analytics | sophistication {*none, minimal, intermediate, advanced, expert, innovator, strategic*}, primary_motivation {*dominance, ideology, notoriety, personal-gain, personal-satisfaction, unpredictable*} | |
| Thread-level analytics | name, description, aliases, roles {*independent, infrastructure-architect, infrastructure-operator, malware-author*} | first_observed, last_observed |
| Forum-level analytics | threat_actor_types {*activist, criminal, hacker, sensationalist, terrorist*}, resource_level {*individual, club, contest, team*} | number_observed, objects {*Artifact, Domain Name, Email Address, File, IPv4, IPv6, MAC Address, Software, URL*}, object_refs |

Furthermore, the structure of the presentation component consists of three independent modules. Each module handles an object class and supports create, read, update, delete (CRUD) functionalities regarding the management of STIX objects belonging to it. The following actions summarize the modules' functionality: (a) create an object if the analytics layer provides new data, (b) read STIX objects from the database if requested, (c) update STIX objects if new data are related to them, and (d) delete STIX objects if requested. Besides the above functionality, the *observables* module manages the STIX observables produced by the aggregation component of the processing layer (see Section 5.3.2). Figure 8 depicts the structure of the presentation component.

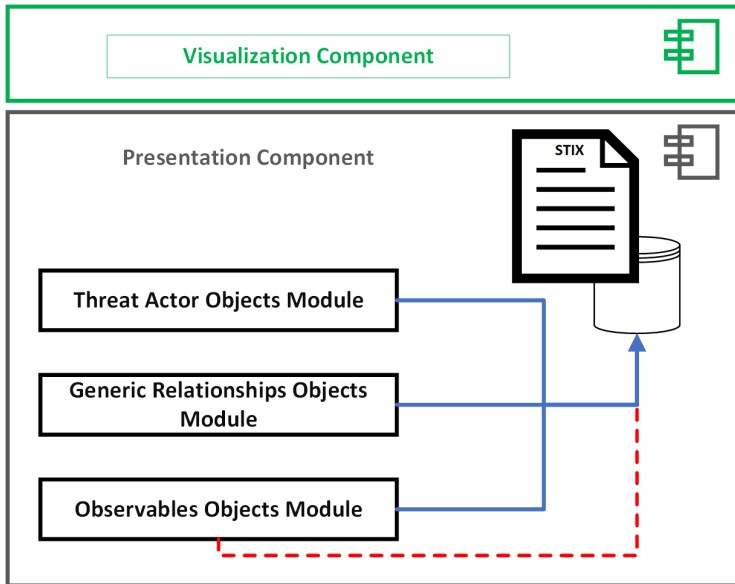

**Figure 8.** SECDFAN Presentation Layer Components.

*5.6. Communication Layer*

The communication layer adds the required capabilities to a CTI system to effectively communicate the results of threat intelligence (CTI products) with other systems. In the case of SECDFAN, we examined if the system was expected to communicate with others and if the functions of this layer were required. Table 8 summarizes SECDFAN's requirements and the communication layer component.

**Table 8.** Application of CTI Reference Model: Communication Layer.

| Function | Req. (Yes/No) | Component | Layer Component |
|---|---|---|---|
| CTI Products Exchange | Yes | Exchange | Communication |
| Privacy protection | No | - | |

SECDFAN aims to share the created CTI products. As a result, the communication layer adds a critical capability to the system. We represented the capability of handling and sharing CTI products with a communication component on the SECDFAN's reference architecture (see Figure 9). In addition, SECDFAN must be capable of participating in information-sharing communities, so the CTI products exchange function is required and represented with the exchange component in its reference architecture. The implementation of this component is expected to be based on TAXII server [37]. Furthermore, there is no need for privacy protection functionality since SECDFAN uses data from publicly available sources.

*5.7. Quality Control Layer*

The quality control layer deals with the management of quality regarding a CTI system, irrespective of whether this quality refers to the CTI system itself or the CTI products. Since SECDFAN is designed to create CTI products following both manual and automatic analytics approaches (see Section 5.4), all the functions and the capabilities of this layer are required. Table 9 summarizes those requirements and their respective components.

**Table 9.** Application of CTI Reference Model: Quality Control Layer

| Function | Req. (Yes/No) | Component | Layer Component |
|---|---|---|---|
| Feedback collection | Yes | Feedback collection | Quality control |
| Quality metrics calculation | Yes | Metrics calculation | |
| CTI products evaluation | Yes | Evaluation | |

The components of this layer are presented in Figure 9. The quality control component manages all the quality requirements within SECDFAN (e.g., performance, CTI product's quality, efficiency). The feedback collection component manages the expert's feedback regarding the produced CTI products. The metrics calculation component combines all the required information and calculates the quality metrics. Finally, the evaluation component assesses the overall quality of the created CTI products.

*5.8. Collaboration Layer*

The collaboration layer deals with the capabilities of a CTI system to support the management of CTI operations and the collaboration of security experts. Since SECDFAN is purpose-specific (i.e., extraction of CTI from DFs), there is no requirement for CTI operations management. However, SECDFAN should support collaboration between analysts and the integration of other collaboration tools. Those requirements and their respective reference architecture components are presented in Table 10.

**Table 10.** Application of CTI Reference Model: Collaboration Layer.

| Function | Req. (Yes/No) | Component | Layer Component |
|---|---|---|---|
| CTI operations planning | No | - | Collaboration |
| Analysts collaboration | Yes | Analysts collaboration | |

In Figure 9, we present two components for this layer. The collaboration component handles the integration of external tools into SECDFAN. At the same time, analysts' collaboration handles the collaboration between security analysts working with SECDFAN and provides their feedback about the CTI products to the feedback collection component.

*5.9. SECDFAN Reference Architecture*

Having applied the CTI reference model, we developed the SECDFAN reference architecture, presented in Figure 9. The SECDFAN reference architecture consists of layers' and functions' components and lines that illustrate the information flow between those components. There are three types of lines: the *arrow* line, the *information control* line, and the *information select* line. The arrow line represents an information flow that compulsorily passes from one component to another. The information control line indicates that a component controls what information flow passes to another component. Finally, the information select line indicates that a component can control the information flow passed by another component.

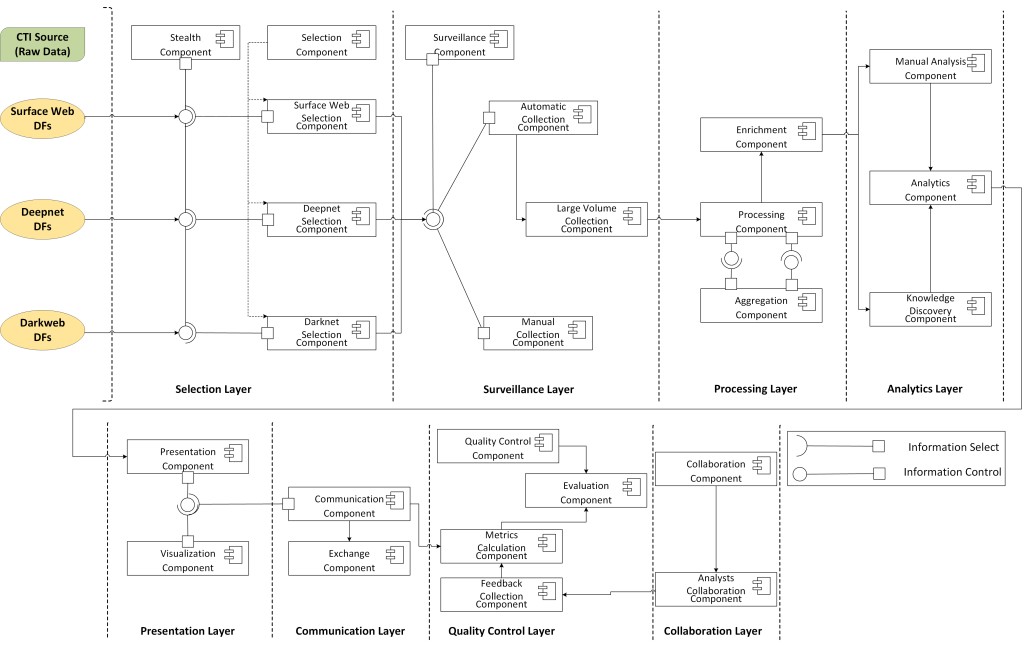

**Figure 9.** SECDFAN Reference Architecture.

*5.10. Comparison with Proposed Systems*

The produced SECDFAN reference architecture is the blueprint for developing the SECDFAN system, which is part of our future work. Consequently, ordinary validation and verification methods (e.g., using a set of text data collected from discussion forums) are not applicable. Instead, we compare the components of the SECDFAN reference architecture with those of other systems in Section 2.4, thus highlighting the limitations of those systems in relation to the complete set of CTI issues. We summarize this comparison in Table 11. We used the eight layers of the CTI reference model [2] and we inserted a plus sign for each system with at least one component in its architecture, handling a function of the respective layer.

**Table 11.** Comparison of Architectures of CTI Systems.

| Layer/CTI System | SECDFAN | iACE | NEDe-Tector | Vulcan | [21] | INTIME | [23] |
|---|---|---|---|---|---|---|---|
| Selection | + | | | | | + | |
| Surveillance | + | + | + | + | | + | + |
| Processing | + | + | + | + | + | + | + |
| Analytics | + | + | + | + | + | + | + |
| Presentation | + | + | | | | + | |
| Communication | + | | | | | + | |
| Quality control | + | | | | | | |
| Collaboration | + | | | | | | |

## 6. Conclusions

In this paper, we focusd on utilizing discussion forums as a source of CTI. As a result, we proposed SECDFAN and provided a detailed reference architecture for its development after analyzing the requirements for using discussion forums in CTI.

SECDFAN resulted from a systematic and holistic design methodology based on a CTI reference model. SECDFAN modeled all the functionalities and capabilities of a CTI system that deals with the utilization of discussion forums. More specifically, it modeled the functionalities and capabilities of a CTI system, starting from the selection of a data source to the sharing of the CTI products and collaboration of the analysts.

This paper's research contribution and innovative characteristics are summarized in the following points:

- It analyzed the structure of discussion forums, the roles of their participants, and their unique linguistic characteristics in relation to CTI.
- It proposed a semantic schema for the representation of raw data coming from DFs
- It provided a DFs' characteristics comparison based on their origin.
- It designed SECDFAN and explained how different DFs analysis methodologies could be combined into a system for the benefit of CTI.
- It proposed an analysis of DFs based on their semantic representation and not only in the text data.

The main contribution of this work was the development of the SECDFAN reference architecture, which addressed, by design, all issues related to the utilization of discussion forums for CTI product generation. This was achieved by following a holistic approach that ranged from the selection of discussion forums to the collaboration of security analysts.

Finally, our future research work aims to use the SECDFAN reference architecture as the blueprint for the development of the SECDFAN system. Moreover, our short-term aims include the development of a CTI-based automatic annotation component for DFs' threads and a component for measuring and improving the quality of DFs' raw data and CTI products.

**Author Contributions:** Conceptualization, G.S. and P.F.; methodology, G.S.; validation, G.S., P.F. and I.M.; formal analysis, G.S.; investigation, G.S.; data curation, G.S.; writing—original draft preparation, G.S.; writing—review and editing, G.S, P.F. and I.M.; visualization, G.S, P.F. and I.M.; supervision, P.F. and I.M. All authors have read and agreed to the published version of the manuscript.

**Funding:** This research received no external funding.

**Informed Consent Statement:** Not applicable.

**Data Availability Statement:** No new data were created for this research.

**Conflicts of Interest:** The authors declare no conflict of interest.

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
