# Peer review of "SECDFAN: A Cyber Threat Intelligence System for Discussion Forums Utilization"

_2673-4117, doi:10.3390/eng4010037_

Round 1

Reviewer 1 Report

This article analyzes the structure and unique characteristics of discussion forums and its CTI data. The SECDFAN system proposed in this paper is customized for discussion forums. It was based on the CTI reference model and the requirements analysis method, which were published by the author in Electronics. SECDFAN models the functionalities and capabilities of a CTI system, starting from the selection of a data source to the sharing of the CTI Products and collaboration of the analysts, which is worthy of recognition.

Strengths

+ Analyzed the difference in linguistic characteristics between discussion forums and real-time chat.

+ The functions and definitions of the 8-layer reference model are detailed and effective.

Minors

- In section 3.2.2, the description of Typical Thread Structure is too abstract and lacks examples.

- In line 244, you did not indicate the source of these 30 discussion forums, are they representative?

- In line 311, you should be referencing Table 3 instead of Table 2.

- Some terms are not defined when they first appear, such as CMC in line 105, TSF in line 327.

Author Response

Response to Reviewer 1 Comments

Point 1: In section 3.2.2, the description of Typical Thread Structure is too abstract and lacks examples.

Response 1: Thank you for the remark. You obviously meant section 4.1.1. We have added in the respective section the following paragraph to clarify the typical thread structure immediately after Figure 4.

“To clarify the typical thread structure, we use the data collected from a real dark web discussion forum, as an example. In that case, a thread's initial post (e.g., GENERAL FEATURES: --2500 lines of code...) was from a user who was selling a new malware. This post triggered an initial discussion where other users replied directly to the initial post by posting new messages (e.g., “nice will try”, “do you recommend a good crypter?”). However, in some of those replies, the thread's initiator or other users decided to reply (e.g., “well I would vouch for mine”, “give it a try...”) and create a new discussion. Thus, the typical thread structure can describe and organize these discussions' structure and interactions.”

Point 2: In line 244, you did not indicate the source of these 30 discussion forums, are they representative?

Response 1: We have explained the way how those discussion forums have been selected and their level of representativeness by adding the following at the referred point:

“the authors have selected these forums by utilizing publicly available sources (e.g., Hacking websites, Wikipedia). Since we can not determine the dark web's size and content, we cannot claim that those discussion forums are completely representative. however, this analysis offers a general impression of their characteristics which, to the best of our knowledge, has not been presented in another work.”

Point 3: In line 311, you should be referencing Table 3 instead of Table 2.

Response 3: Thank you. We have corrected the reference.

Point 4: Some terms are not defined when they first appear, such as CMC in line 105, TSF in line 327.

Response 4: We have defined the term CMC at the beginning of section 2.3 and the term TSF in the respective line.

Reviewer 2 Report

Title:  A Cyber Threat Intelligence System for Discussion Forums Utilization

The paper is well-written and organized and the content has good overall merits. However, some important elements should be present before the final acceptance for publication.

Firstly, the motivation for conducting such research is missing. The gap that this research is trying to solve is not presented.  Authors should clearly describe the limitations of the existing solutions that motivate them to conduct such a study. The contribution should be highlighted in the abstract, introduction, and conclusion sections.

Secondly, in the related work, authors should highlight the existing works' limitations and explain the issues that motivate conducting this research.

Thirdly, the paper missed out on the validation and evaluation. What are the outcomes of the designed model? How the proposed system is validated and evaluated.  A comparison between the proposed system and other related work is important.

Finally, the limitations of the proposed system should be highlighted in future works in the conclusion section.

Author Response

Response to Reviewer 2 Comments

Point 1: Firstly, the motivation for conducting such research is missing. The gap that this research is trying to solve is not presented.  Authors should clearly describe the limitations of the existing solutions that motivate them to conduct such a study. The contribution should be highlighted in the abstract, introduction, and conclusion sections.

Response 1: We have added the following paragraph in the introduction to answer the comment about the missing motivation :

“This paper focuses on the utilization of discussion forums from CTI systems in creating CTI products [ 2]. Our motivation originates from the fact that the CTI systems proposed in the bibliography deal with certain but not all aspects of the CTI process [2] that utilizinges discussion forums.”

To describe the limitations of existing proposals and our motivation, we have added a new section in the related work, as presented in response to point 2, and we have slightly modified the last paragraph of the introduction as follows:

“a fourfold review of related works is presented. We outline the works on CTI system modeling (2.1), discuss CTI extraction from text sources (2.2), discuss works of Computer-Mediated Conversations (CMC) and discussion forums (DF) analysis (2.3), and highlight the limitations of the works dealing with CTI systems for discussion forums (2.4).”

The contribution of this work has been highlighted in the abstract, introduction, and conclusion sections by adding the following parts, respectively:

Abstract:

“The contribution of this paper is the development of a CTI reference architecture of a system that, by design, handles all CTI-related issues in creating CTI products by analyzing the content of discussion forums.”

Introduction:

“The reference architecture of the SECDFAN system constitutes the main contribution of this paper since it addresses, by design, all issues related to the creation of CTI products by employing content analysis of discussion forums.”

Conclusion:

“The main contribution of this work is the development of the SECDFAN reference architecture, which, addresses, by design, all issues related to the utilization of discussion forums for CTI product generation. This is achieved by following a holistic approach that ranges from the selection of discussion forums to the collaboration of security analysts.”

Point 2: Secondly, in the related work, authors should highlight the existing works' limitations and explain the issues that motivate conducting this research.

Response 2:  We have added the following section in the related works to highlight the limitations of the existing works and explain our motivations:

“2.4. CTI Systems for Discussion Forums

Collecting and analyzing data from websites in the darknet, deepnet, and surface web for cybersecurity is the subject of many works. Here, we present several of them and discuss their limitations and issues, which motivated us in this work. Liao et al. [20 ] propose the iACE solution for automatic extraction of Indicators of Compromise (IoC)s from security articles published in blogs. However, they only deal with some of the CTI challenges (e.g., source selection, stealthiness) and they use only the content of the articles; not the information regarding the bloggers themselves.

Li et al. [21 ] have developed the NEDetector system to extract neologisms from hacking forums. However, they mainly focus on the analytic parts of the system and do not handle issues related, for example, to data collection and selection of the sources.

Jo et al. [6 ] propose the Vulcan CTI system for extracting CTI data from unstructured text. However, they focus only on the analytics part of their system and the semantic processing of the text.

Similarly, Deliu et al. [22] have designed a CTI system in the form of a model that automatically searches hackers’ posts and extracts CTI. But, again, they mainly focus on the analytics part of their model and do not provide details, for example, of the quality measurement.

Koloveas et al. [23] have proposed the INTIME, a machine learning-based framework that combines several tools and services in one architecture to support security analysts in dealing with various data sources in the process of developing CTI products. But the proposed architecture is not based on a reference model. Consequently, several gaps appear in how INTIME handles some CTI-related issues, like the quality of the refined CTI products.

Finally, Sapienza et al. [ 24] have proposed a CTI system that uses data from cybersecurity experts and forums to generate CTI and to alert cybersecurity specialists on an imminent attack. However, the designers of the proposed CTI systems have focused only on specific components which deal with specific CTI issues, like analytics.

To summarize, we observe a general tendency for the architecture of the proposed CTI systems to focus only on specific parts and CTI-related issues, resulting from the fact that these authors did not follow a reference model during their design. This limitation motivated us to develop a reference architecture for systems that process discussion forums data as a source to CTI, by utilizing the reference model we proposed in [2].”

Point 3: Thirdly, the paper missed out on the validation and evaluation. What are the outcomes of the designed model? How the proposed system is validated and evaluated. A comparison between the proposed system and other related work is important.

Response 3:  We have added the following section to answer the questions of Point 3:

“5.10. Comparison with proposed systems

The produced SECDFAN reference architecture is the blueprint for developing the SECDFAN system, which is part of our future work. Consequently, ordinary validation and verification methods (e.g., using a set of text data collected from discussion forums) are not applicable. Instead, we compare the components of the SECDFAN reference architecture with those of the other systems in section 2.4, thus highlighting the limitations of those systems in relation to the complete set of CTI issues.We summarize this comparison in Table 11. We use the eight layers of the CTI reference model [2]and we insert a plus sign for each system  with at least one component in its architecture, handling a function of the respective layer.”

Point 4: the limitations of the proposed system should be highlighted in future works in the conclusion section.

Response 4:  We have added the following paragraph in future work to highlight the limitations of the proposed reference architecture:

“Finally, our future research work aims to use the SECDFAN reference architecture as the blueprint for the development of the SECDFAN system. Moreover, our short-term aims include the development of a CTI-based automatic annotation component for DF's threads and a component for measuring and improving the quality of DFs raw data and CTI products.”

Reviewer 3 Report

The authors propose a semantic schema for the presentation of data collected from discussion forums. Then, a systematic methodology is applied to design the reference architecture of SECDFAN system, which handles the creation of CTI products. SECDFAN reference architecture is the final product of this work. The paper in general is well-written. The literature review is well-structured and written as well. The research methodology is well-explained and applied.

In line 274: SEDFCAN -> SECDFAN

In line 356: Analytic layer -> Analytics layer

In line 413: message’stime -> add a space between the two words

Author Response

Response to Reviewer 3 Comments

Point 1: In line 274: SEDFCAN -> SECDFAN

Response 1:  We have corrected the misspelling.

Point 2: In line 356: Analytic layer -> Analytics layer

Response 2:  We have corrected the misspelling.

Point 3: In line 413: message’stime -> add a space between the two words

Response 3:  We have added the space.
